# Bayesian estimation of predator diet composition from fatty acids and stable isotopes

Philipp Neubauer[1] and Olaf P. Jensen[2]

[1] Dragonfly Science, Wellington, New Zealand
[2] Department of Marine and Coastal Science, Rutgers University, Institute of Marine & Coastal Sciences, New Brunswick, NJ, USA

## ABSTRACT

Quantitative analysis of stable isotopes (SI) and, more recently, fatty acid profiles (FAP) are useful and complementary tools for estimating the relative contribution of different prey items in the diet of a predator. The combination of these two approaches, however, has thus far been limited and qualitative. We propose a mixing model for FAP that follows the Bayesian machinery employed in state-of-the-art mixing models for SI. This framework provides both point estimates and probability distributions for individual and population level diet proportions. Where fat content and conversion coefficients are available, they can be used to improve diet estimates. This model can be explicitly integrated with analogous models for SI to increase resolution and clarify predator–prey relationships. We apply our model to simulated data and an experimental dataset that allows us to illustrate modeling strategies and demonstrate model performance. Our methods are provided as an open source software package for the statistical computing environment R.

## INTRODUCTION

Quantitative estimates of an animal's diet are a critical component of predator–prey studies, ecosystem models, and ecosystem-based management. Existing methods for estimating diet proportions all have strengths and weaknesses (*Bowen & Iverson, 2012*). Traditional stomach content and fecal matter analysis represent a brief snapshot of diet at a particular place and time and can be invasive, time-consuming, and potentially biased by differential rates of digestion of prey or egestion of identifiable prey parts (*Bowen & Iverson, 2012*). Chemical markers such as stable isotopes (SI) and fatty acid (FA) profiles solve some of these problems. For example, both approaches integrate diet composition over an extended time period—typically weeks to months, depending on tissue turnover rates (*Tucker, Bowen & Iverson, 2008*). These advantages have led to rapid growth in the use of chemical markers in diet studies (*Elsdon, 2010*; *Williams & Buck, 2010*; *Kelly & Scheibling, 2011*). However, chemical dietary markers often lack the specificity of traditional stomach content analysis. In particular, several prey species often have similar

Corresponding author
Philipp Neubauer,
neubauer.phil@gmail.com

isotopic signatures. More recent studies have sought greater dietary resolution through the use of SI of other elements in addition to carbon and nitrogen (e.g., *Belicka et al., 2012*), compound specific SI ratios (e.g., *Budge et al., 2008*; *Jack & Wing, 2011*), or a combination of stomach content analysis and SI or FA profiles (e.g., *Pethybridge et al., 2012*). The use of SI and FA profiles in combination also holds great promise; however, studies that have used both chemical markers have been qualitative (e.g., *Guest et al., 2009*) or based on positive correlation of results from both methods (*Tucker, Bowen & Iverson, 2008*).

Analysis tools for SI data have become very sophisticated in recent years, starting with the development of general Bayesian analysis tools for estimating diet proportions, and leading to customized (hierarchical) models for individual applications (*Moore & Semmens, 2008*; *Hopkins & Ferguson, 2012*; *Parnell et al., 2013*). The latter models can, for instance, estimate dietary differences of geographically distinct populations (*Semmens et al., 2009*), accommodate temporal changes in diets or estimate the effect of covariates (e.g., age, size, sex) on diet proportions (*Parnell et al., 2013*). While these models provide a considerable step towards ecologically relevant models in diet studies, the underlying SI data is limited in the resolution that it can provide. Since typically only 2–3 SI are measured, the contrast that is achievable from such a low number of variables is necessarily limited, especially when the number of potential prey items increases (*Phillips & Gregg, 2003*; *Ward et al., 2011*). In particular, when a system contains more potential prey items than SI markers, the system becomes under-determined, leading to potentially biased estimates of diet composition (*Phillips & Gregg, 2003*; *Fry, 2013a*; *Semmens et al., 2013*; *Fry, 2013b*; *Brett, 2014*). Optimally aggregating prey items into prey groups may circumvent this problem (*Ward et al., 2011*), but may also be less satisfactory in complex food webs.

FA profiles can, in theory, provide considerably more resolution compared to SI data, due to the large number of potential fatty acids that can be measured. However, the data structure of FA profiles is quite different to that of SI, because measured FA proportions that make up the profile are constrained to sum to one. Direct adaptions of mixture model methodologies developed for SI to FA data so far ignore this constraint (e.g., *Galloway et al., 2014a*; *Galloway et al., 2014b*) and may therefore give biased estimates (*Aitchison, 1982*). *Blanchard (2011)* developed a Bayesian mixing model for diet inference from FAs that accounts for the compositional constraints on FA data (furthering the development of Bayesian mixing models for compositional data by *Billheimer (2001)*), showing that model based inferences of predator diets from FAs are achievable. Nevertheless, most studies employing FA profiles remain either qualitative in their estimates of prey proportions in predator diets, or use Quantitative Fatty Acid Signature Analysis (QFASA; *Iverson et al., 2004*) to obtain quantitative estimates of diet proportions.

QFASA is the only purpose built method thus far for use with FA profiles, and, in contrast to recent (Bayesian) SI and FA mixing models, relies on a distance metric rather than a model based formulation to estimate the most likely diet proportions. This framework provided the first quantitative approach to estimating diet proportions using FAs and it has already seen widespread use, particularly in studies of marine mammals (*Bowen & Iverson, 2012*) and seabirds (*Williams & Buck, 2010*). Nevertheless,

QFASA has a number of limitations. Since it is not based on a probabilistic model, it is difficult to estimate the contribution of multiple sources of uncertainty associated with estimated diet proportions (but see *Stewart & Field, 2011*; *Stewart, 2013*, for a treatment of confidence intervals in QFASA). The absence of an explicit model also makes it impossible to build ecological mechanisms (e.g., covariates of consumed diets) directly into the model. Furthermore, uncertainty about conversion coefficients representing enrichment and preferential uptake of FAs cannot be considered, yet small changes in these coefficients can lead to differences in inferred diet proportions (*Wang, Hollmen & Iverson, 2010*).

Given the discrepancy in methods applied to SI and FA profiles, it is perhaps not surprising that their joint application has commonly relied on qualitative comparisons. Because both markers integrate diet composition over often comparable time-scales, however, an explicit integration of these data types could provide substantial benefits. While FA profiles could mitigate the resolution problem in SI data, SI data could provide increased resolution and clarify predator–prey relationships, the knowledge of which is usually a pre-requisite for FA profiles. For example, for many non-modified fatty acids, FA profiles alone cannot discriminate between the case of two species which share a common diet and the situation in which one of these species eats the other. In either case, the two species may have similar FA profiles. The addition of a stable isotope with trophic fractionation (e.g., $^{15}N$), however, can readily distinguish predation from dietary overlap.

Here, we develop a mixing model for FA profiles based on a probabilistic model whose parameters are estimated using Bayesian methods, and explicitly integrate SI in the estimation of diet proportions. Using both simulated and experimental data, we highlight the potential benefit of explicit integration of FA with SI data to increase the precision of diet estimates.

## METHODS

### A Bayesian mixing model for fatty acid profiles

Bayesian models for SI data are commonly based on the assumption that SI ratios are normally distributed. This assumption cannot be made for FA profiles, since for most methods of analysis, the concentration of individual FAs is normalized to the total lipid content of the sample. Thus, the FA profiles are a collection of proportions (referred to as a composition), which lie between 0 and 1, and are constrained to sum to 1. A common solution to this problem, however, is to consider transformations that make the data approximately normal (*Budge, Iverson & Koopman, 2006*). To construct our model, we considered the additive log ratio transformation (*Aitchison & Bacon-Shone, 1999*), also called alr transformation, such that

$$y_{i,s} = alr(\phi_{i,s}) = log\left(\frac{\phi_{i,s,1...p-1}}{\phi_{i,s,p}}\right) \tag{1}$$

where $\phi_{i,s}$ is the $p$-variate fatty acid composition of individual $i$ of prey species $s$, with a total of $n$ potential prey species considered. Note that in the following we often drop the subscript for FAs, e.g., $\phi_{i,s}$ and $y_{i,s}$ are thus $p$ and $p - 1$ dimensional vectors, respectively.

We assumed that the distribution of $y$ is multivariate normal, with species specific mean $\mu_s$ and covariance matrix $\Sigma_s$, or $y_{i,s} \sim N(\mu_s, \Sigma_s)$. A vaguely informative prior on $\mu_s$ and $\Sigma_s$ allows for uncertainty in prey distributions to propagate to estimates of diet proportions (*Ward, Semmens & Schindler, 2010*).

Each prey source represents a proportion $\pi_j$ of the diet of predator $j$, and analogous to stable isotope mixing models, predator FA profiles are then a linear combination of prey FA profiles, normalized to sum to one. Since predators may selectively assimilate or metabolize FAs (*Iverson et al., 2004*; *Budge, Iverson & Koopman, 2006*; *Rosen & Tollit, 2012*), we specified prey-specific conversion coefficients $\kappa_s = \kappa_{s,1} \dots \kappa_{s,p}$ for each of the $p$ FAs (*Rosen & Tollit, 2012*). Furthermore, the $n$ prey species likely have different fat content $\Phi$ that will affect the total amount of FAs assimilated from each prey species by the predator. The expected FA profile of predator $\tau_j$ is then a linear combination of the prey FA profiles, modified by conversion coefficients for each fatty acid $p$ and fat content for each prey $i$:

$$t_j \sim N(alr(\tau_j), \Sigma_\tau) \tag{2}$$

$$\tau_j = C\left\{ \sum_s^n (\pi_{j,s}\Phi_s)(\kappa_s \otimes \phi_{j,s}) \right\}. \tag{3}$$

Here, $C$ is the closure operation which normalizes the FA profiles to sum to one and $\otimes$ is the outer (element wise) product. $\phi_{s,j}$ is the FA profile of prey items of species $s$ consumed by predator $j$. Similarly to *Parnell et al. (2013)*, we thus assumed that individual predators do not necessarily feed on 'average' prey items, but rather consume prey items with signatures drawn from the estimated prey distribution. We formulated predator signatures $t$ as draws from a normal distribution after transformation. We further assumed that $\Phi$ and $\kappa$ are log-normally and gamma distributed, respectively, around known mean and variance values (estimated or calculated from controlled diet experiments, see below). The closure operation in Eq. (3) (i.e., the sum-to-one constraint on the FA profiles) leads to $\kappa$ being determined in terms of relative uptake of FAs (i.e., up to a multiplicative constant), and implicitly makes the multivariate prior distribution over all $\kappa$ a Dirichlet distribution. The same logic applies to $\Phi$, and in both cases we opted for formulations that can be readily parametrised from sample means and variances from controlled diet experiments.

The diet proportions $\pi$ of predators are the main focus of investigation in diet studies. These may be modeled at the population level (thus dropping the subscript $j$ in expressions (2) and (3)) or at the individual level, as suggested in expressions (2) and (3). In the latter case, individual predator FA profiles can be modeled as random samples from a population level distribution of predator diet proportions. Recent approaches to stable isotope mixing have focused on transformations of the diet proportion vector $\pi$ to get around the problems associated with the compositional nature of the diet proportions in such a hierarchical setup, and we followed this approach in our model. The diet proportions were transformed using centered log-ratio (clr) transformations (*Semmens et al., 2009*), such that the support is the real line rather than the interval [0;1], and we then assumed that $clr(\pi_j) \sim N(\Pi, \Sigma_\Pi)$, where $\Pi$ is the vector of mean (population level) diet proportions.

It is then possible to model diet proportions as a function of covariates, such as size, sex, or region (i.e., in a regression formulation, *Parnell et al., 2013*). While this approach is appealing, it adds to computation time needed to estimate model parameters, and often results in slower convergence. We therefore use a vague Dirichlet prior on the proportions when convenient (i.e., when we estimate only population level parameters).

Depending on the number of samples for prey and predators, it may be necessary to use informative priors for $\Sigma_s$ and $\Sigma_\tau$. Both were given inverse-Wishart priors, and since both are co-variances of transformed data, it is not straightforward to formulate default priors for these parameters. We have found that, in practice, manual adjustment of these priors is often needed to be able to achieve convergence and mixing (efficient exploration of the posterior distribution by the sampling algorithm) of the Markov Chain Monte Carlo (MCMC) routine employed to estimate model parameters. This is especially true when there are few source and/or predator samples. Our code (see below) allows for high level adjustment of these parameters through the specification of the order of magnitude of the diagonal of each covariance matrix.

## Joint diet estimation from FA profiles and SI

There are at least three potential benefits of integrating FA profiles and SI data: (i) increased information to discriminate among sources, (ii) the potential of SI to resolve predator prey relationships due to trophic enrichment of SI, and (iii) the potential reduction in estimation error due to a larger body of research on fractionation coefficients for stable isotopes as opposed to conversion coefficients in FA profiles. Integrating the two complimentary types of data in a single model to estimate diet proportions may thus considerably improve estimates of diet proportions over estimation from either data-source alone.

Our model for FA profiles is conceptually similar to recent models proposed for SI data, and integration of FA profiles and SI data into a single model is straightforward in the present setting. We assumed that the vector of SI signatures of sampled prey items $q$ follow a multivariate normal distribution, such that $y_{q,s}^{SI} \sim N(\mu_s^{SI}, \Sigma_s^{SI})$, where the superscript *SI* denotes that these are stable isotope signatures. Predator SI signatures are again a linear combination of prey SI, this time modified by additive fractionation coefficients $\gamma$. Fractionation may, in turn, depend on prey isotope concentrations (*Hussey et al., 2014*; *Caut, Angulo & Courchamp, 2009*). In our model, we assumed additive fractionation, and suggest that concentration dependence is taken into account when specifying distributions for prey and SI specific fractionation coefficients $\gamma_s$ (see examples below). The expected SI signature for predator $r$ is then

$$t_r^{SI} = \sum_s^n \pi_{r,s} \left( y_{q,r} + \gamma_s \right) \tag{4}$$

$$clr(\pi_r) \sim N(\Pi, \Sigma_\Pi) \tag{5}$$

$$\gamma_{s,SI} \sim N(\nu_{SI}, \sigma_{SI}). \tag{6}$$

Note that the different subscripts to the FA profile model imply that there is no need to have SI and FA profiles from the same prey or predator samples, as long as we can assume that the prey samples are drawn from the same statistical population as those for FA profiles, and that individual diet proportions of predators are drawn from the same population distribution of diet proportions.

The exact formulation of the integration of SI and FA profiles depends on the assumptions that one is comfortable with in a given setting: identical dietary proportions may be appropriate if diets (and hence SI and FA profiles) are thought to be stable, or if both chemical tracers are thought to integrate over similar time-scales. If the time scales of these two elements are thought to be different (e.g., for different tissue types), individual diet proportions may be more appropriate, and may be drawn from an overall population distribution of diet proportions.

An R (*R Core Team, 2014*) package (called fastinR) implementing methods outlined here, along with simulated examples and the analysis of experimental data described further below, is available on the open source repository github.com/philipp-neubauer/fastinR. Models implemented in the package include the above-mentioned formulations for population level diet estimates, individual diet estimates as well as linear model (regression and ANOVA) formulations for diet proportions, all available for SI and FA profiles individually or as combined models (see below). Model parameters were estimated using MCMC methods implemented in JAGS (*Plummer, 2003*), called from R through higher level functions in the fastinR package that allow for data input, inspection and manipulation.

## Simulation studies

We initially explored the feasibility and performance of our model setup in a range of simulations, which are illustrated (including code) in Supplemental Information 1. The initial feasibility simulations used a set of three potantial prey species (30 samples per species) with two stable isotopes (i.e., an under-determined system) and 12 fatty acids. To clearly illustrate the method itself, prey-source separation was chosen to readily discriminate prey items for both markers, but with enough variability in prey profiles to illustrate uncertainty propagating through to diet estimates.

Simulations were also used to explore sensitivities of inferred diet proportions to the source configuration and diet evenness in a series of simulation experiments, using 10 simulated fatty acids and four potential prey items. We hypothesized that estimated diet proportions are sensitive to diet source separation in FA profile space, co-linearity in FA profile space (*Blanchard, 2011*) and diet makeup (e.g., specialist versus generalist diets). Further details and simulation results can be found in Supplemental Information 2.

## Selecting fatty acids for analysis: an ordination approach

A potentially large number of FAs are available from analysis methods such as gas-chromatography. A common practice is to simply set a threshold and keep the most abundant FAs for analysis. This practice may, however, discard potential useful information, and a more judicious approach is to retain FAs based on the among diet

source variability that they explain. *Wang, Hollmen & Iverson (2010)* used a method by which they tested the QFASA method on a series of subsets to determine the subset that gave the best accuracy. Although feasible, such a method can be time consuming with fully Bayesian models, which can take a long time to run with a large dataset.

Here, we propose a variable selection method based on constrained ordination, which considers the contribution of individual fatty acids to axes separating diet sources. Based on this contribution relative to the overall separation, the user can choose FAs that contribute most to source separation. This procedure is intended to reduce computation time (and dimensionality) of the models, while retaining as much accuracy in diet estimates as possible. Further details about the procedure are given in Supplemental Information 3.

## Estimating predator diets in a controlled experiment

To illustrate the potential of the models presented above, we analysed data from an experimental study by *Stowasser et al. (2006)*, which investigated changes in squid FA profiles and SI as a function of diet treatments. The treatments consisted of exclusive fish and crustacean diets, as well as switched and mixed diets, with the former switching diets from fish (henceforth SF, $n = 4$) to crustacean (SC, $n = 5$) after 15 days of the 30 day experiment.

In order to apply our model, we first estimated conversion coefficients of FA profiles and fractionation in SI, using squid from the 30 day diet treatments feeding exclusively crustacean and fish diets. The model for estimation of SI fractionation followed the model in *Hussey et al. (2014)*, thus accounting for diet $\delta^{15}N$ and $\delta^{13}C$, and used their results as priors for fractionation parameters for $\delta^{15}N$, and results from *Caut, Angulo & Courchamp (2009)* to construct priors for $\delta^{13}C$. Estimation of FA conversion coefficients used expressions (2) and (3) with proportions assumed known from feeding trials. Computational details on the estimation of conversion coefficients and fractionation are given in Supplemental Information 4.

In our diet analysis, we analyzed samples from the switched diet treatments, and used both SI and FA profiles to investigate the ability of our models to accurately estimate diet proportions in either treatments. We subset the data to use only squid from the switched diet experiment that were analysed for FA profiles and SI after at least 10 days under the respective treatment. We only had overlapping SI and FA profiles for the SC treatment squid, and we therefore started by analyzing this treatment in isolation to demonstrate that both SI and FA profiles can resolve diet proportions, and to demonstrate the benefit of using the two tracers in a joint model. We then analyzed the SF treatment squid, for which we only had 3 specimens with FA profiles and 1 specimen with SI. The markers available for this treatment did not overlap for any of the sampled squid.

Lastly, we estimated individual diet proportions in the SC treatment. To demonstrate how the model based approach to diet estimation can be use to answer ecologically relevant questions about predator diets, we also analyzed SF and SC treatment squid together in a linear model setup that investigated treatment differences explicitly. The linear model used treatment dummy variables to estimate individual intercepts for each treatment and prey

PeerJ ____________________________________________

combination, and allowed us to test for significant differences in diet between treatment groups, conditional on the data and priors.

FA profile analyses used data obtained by analyzing digestive gland tissue, which is thought to rapidly assimilate dietary FAs in relatively unmodified proportions relative to the original diet (e.g., *Phillips, Jackson & Nichols, 2001*). SI were analyzed from muscle tissue since we had more individuals sampled for SI from this tissue, which may be more prone to fractionation and slower turnover than digestive glad tissue. In the original study, a total of 25 FAs were reported. Here, we selected FAs using ordination methods described above. For estimation of model parameters, priors for prey and predator specific variances were adjusted manually to give reasonable behaviour in the MCMC algorithm. The analyses are detailed in Supplemental Information 5.

## RESULTS

### Simulation studies

Simulated test cases suggested that our model can estimate diet proportions from both SI and FA profiles (Supplemental Information 1), with accuracy depending mainly on source separation and diet evenness (Supplemental Information 2). For very uneven diet proportions, such as in the feeding trials analyzed in the squid example, we found the choice of posterior means as point estimate for diet proportions inevitably introduced error at the margins of the 0–1 interval when compared to true simulated diet proportions.

Models with low accuracy conversion coefficients (with prior mean for all FA set to 1 and large prior variance) also performed substantially worse than models with accurately specified coefficients when comparing point estimates of diet proportions to simulated diet proportions (Supplemental Information 2), showing decreasing accuracy with increasing variance among simulated convergence coefficients.

### Estimating predator diets in a controlled experiment

Dimension reduction by NMDS on FA profiles of squid and their potential prey suggested that crustacean diets were readily distinguishable from fish diets (Fig. 1A). For fish diet items, however, no single fish species could be clearly distinguished from any other fish species. Predator signatures of switched diet squid aligned with their respective diets after correcting by posterior means of estimated conversion coefficients. The latter were different from expected ($1/p$) for many FA in the analysis (Supplemental Information 4).

Selection of FAs using constrained ordination lead to four FAs, 22.6n.3, 20.5n.3, 20.4n.6 and 18.1n.9 being retained for analysis (Fig. 2), accounting for a total of 74% of total among source variation on ordination axes while maintaining a low prey matrix condition number ($\kappa = 15.67$), suggesting limited co-linearity. The matrix condition number nearly doubled when the next most important fatty acid ($\kappa = 29.17$) was added and increased exponentially thereafter with the addition of other FAs. The resulting NMDS plot suggested that the reduction from 22 to four FA did not significantly alter the configuration of predators and prey items in FA profile space, despite the drastically lowered number of input dimensions (Fig. 1B). Retaining a larger subset of FAs (8 FAs) did not qualitatively

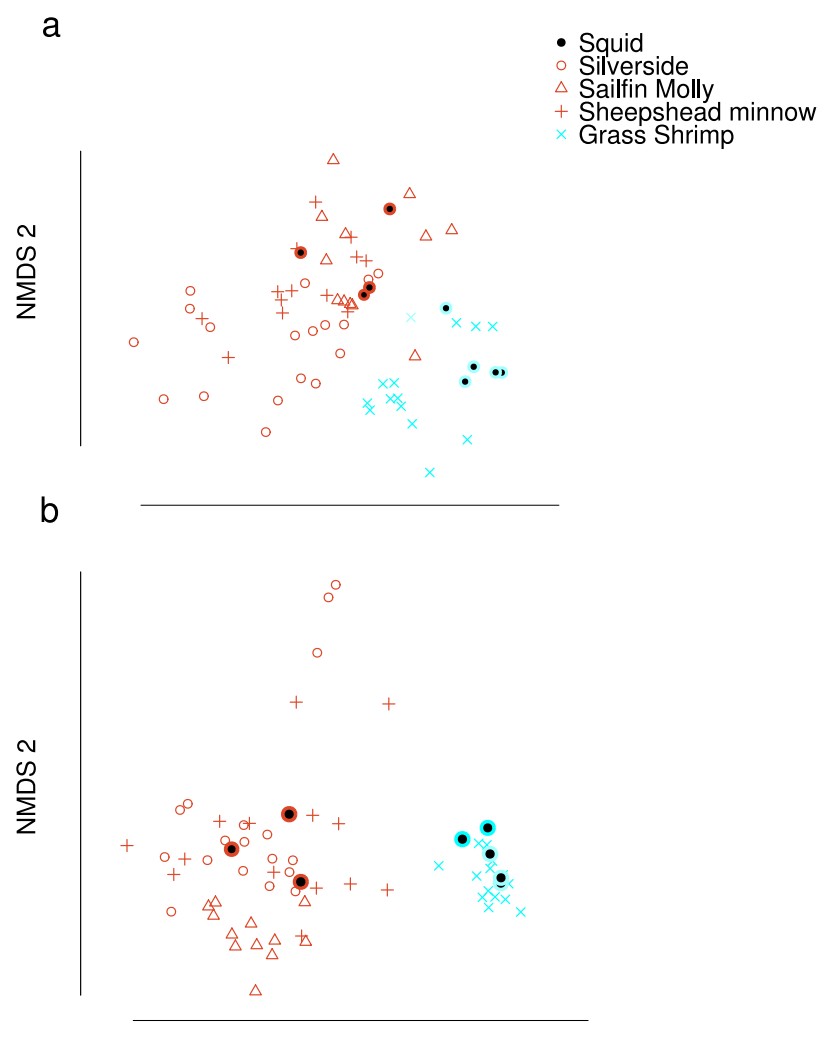

**Figure 1 Non-metric Multi-Dimensional Scaling (NMDS) plots of fatty acid profiles for squid and their potential prey (A) before and (B) after variable selection.** Coloured circles around black points show squid from fish (red) and shrimp (blue) diet treatments. Note that estimated conversion coefficients were applied to scale the data.

alter the results, but did lead to lower uncertainty in diet proportion estimates, suggesting that we lost some relevant information by retaining only four of 25 original FAs to reduce computational requirements.

SI also showed clear separation between crustacean and fish prey (Fig. 3), but showed two groups for fish prey items, both consisting of specimens from more than one fish species. Squid $\delta^{15}N$ was also substantially lower than any of the prey species analysed even after correcting for estimated fractionation coefficients.

FA profiles were able to resolve population level SC treatment squid diets, correctly suggesting a diet predominantly based on crustaceans (Fig. 4). While uncertainty about the exact diet proportions remained for both crustaceans and fish, most of the posterior den-

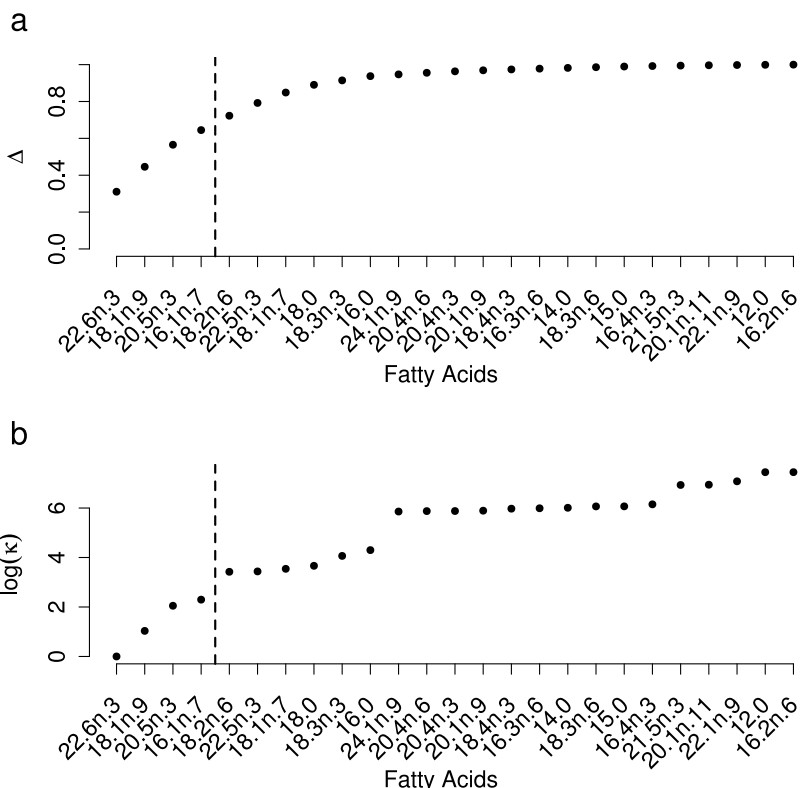

**Figure 2 Selection of a subset of fatty acids for analysis.** (A) Cumulative proportion of between prey variance along constrained analysis of principal coordinates (CAP) axes explained by individual fatty acids being added to the dataset, ordered by the contribution of each fatty acid to the total variance. (B) Prey matrix condition number as a function of individual fatty acids being added as in (A).

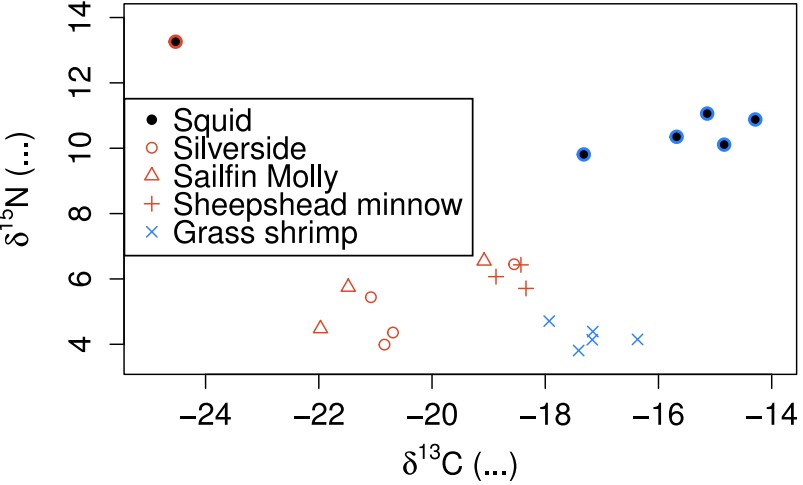

**Figure 3 Stable isotope signatures of squid and their potential prey.** Coloured circles around black points show squid from fish (red) and shrimp (blue) diet treatments.

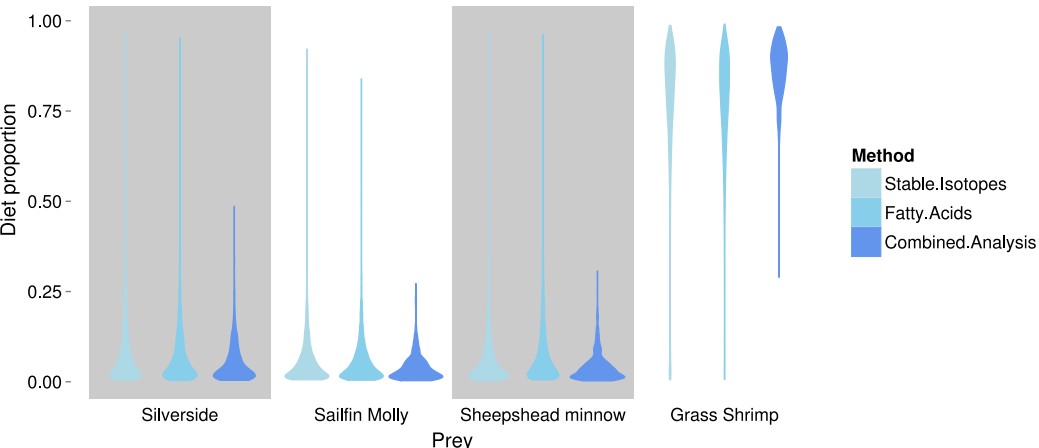

**Figure 4 Posterior densities for diet proportion estimates of SC (crustacean only diet) treatment squid based on fatty acid (FA) profiles, stable isotopes (SI) and a combined (FA & SI) analysis.**

sity for squid diet proportions was clearly concentrated towards high proportions of crustaceans. For fish, posteriors were peaked near zero, however, all fish species posteriors had long tails that spanned nearly the whole interval of possible diet contributions. An analyses based on SI alone gave very similar results, despite different tissue types examined (Fig. 4).

Combining the two markers lead to a substantial reduction in the uncertainty of estimated diet proportions (Fig. 4), and suggested a clear dominance of crustaceans in the diet. For the combined analysis, the spread of the posterior distribution for crustaceans in the squid diet was reduced by approximately 30%, and most of the probability density was shifted closer to one, and the reductions in the spread of posterior distributions for fish diet items were as high as 70%. Lastly, estimates of individual diet proportions closely mirrored population level estimates (Fig. 5).

Due to overlap of fish species in FA profile and SI space, similar models for SF treatment fish were unclear about the contribution of individual fish species (Fig. 6), but suggested that crustaceans were a small part in the diet of these squid. SI and FA profiles combined (i.e., adding one squid with SI but no FA profile data) did not provide much improvement for individual fish species, and the linear model setup was unable to identify significant differences between diet proportions of individual prey items in the two treatments due to uncertainty about individual fish species' contributions (Supplemental Information 5). However, combining fish species post-hoc as the sum of individual posterior distributions clearly shows a fish based diet (Fig. 7).

## DISCUSSION

We presented here a general method for analyzing FA profiles in a Bayesian mixing model, and demonstrated that the method can be used to estimate diet proportions in feeding trials while accounting for fatty acid conversion and diet fat content. The Bayesian framework allows explicit representation of uncertainty about mixing proportions as a function of uncertainty about prey distributions, conversion coefficients and fat content.

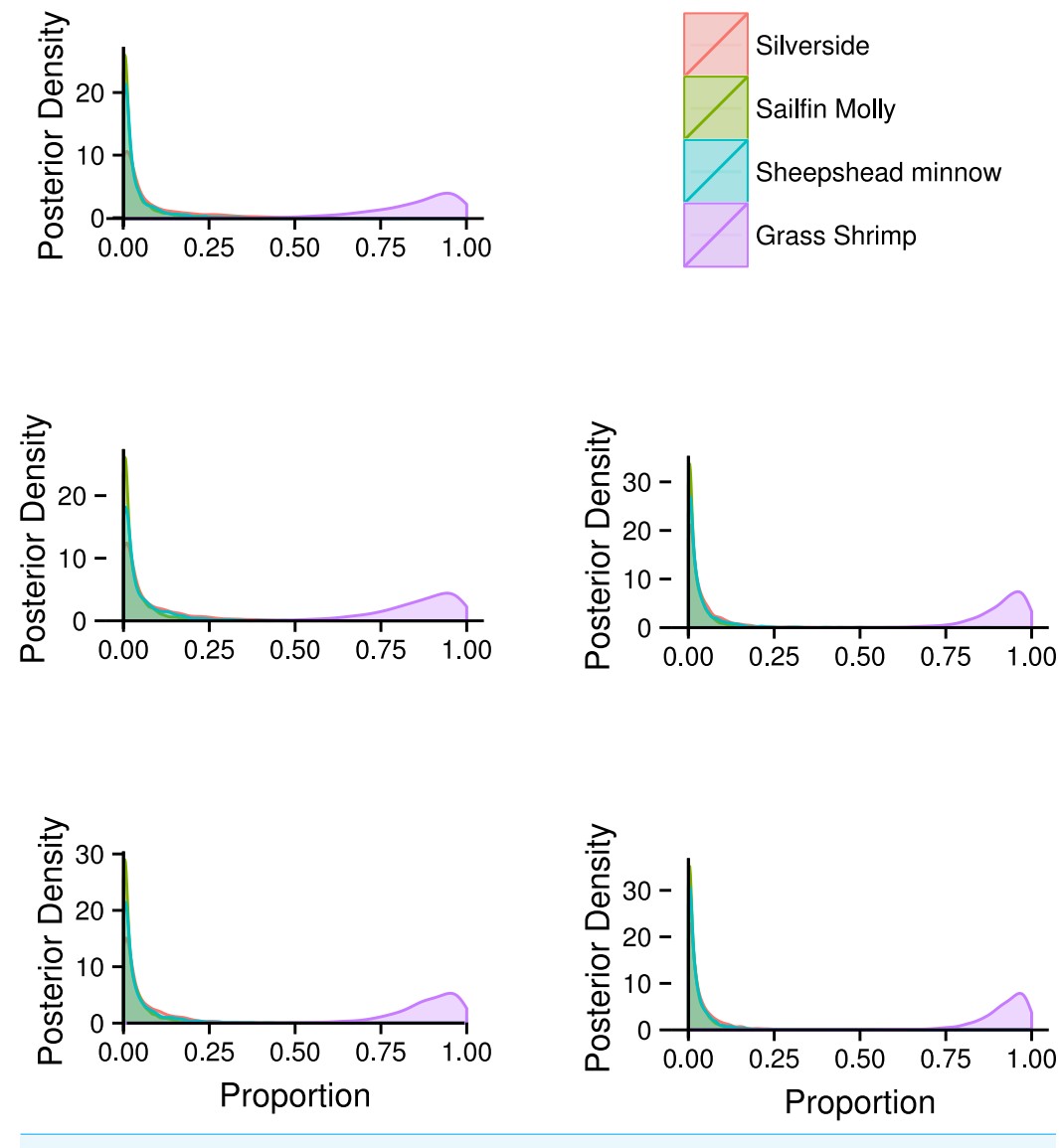

**Figure 5** Posterior densities for individual diet proportion estimates of SC squid based on a hierarchical model for diet proportions using both fatty acid (FA) profiles and stable isotopes (SI).

The general mixing model framework also allowed us to integrate SI and FA profiles into a joint model for diet estimation. Both approaches have their own limits, and the application to squid feeding trials suggests that their combination can substantially reduce uncertainty in diet estimates. As an increasing number of studies combine these two tracers (*Tucker, Bowen & Iverson, 2008*; *Guest et al., 2008*; *Guest et al., 2009*; *Stowasser et al., 2006*; *Van der Bank et al., 2011*; *Jaschinski, Brepohl & Sommer, 2008*), we suggest that a quantitative method to explicitly compare and combine markers will allow practitioners to make more robust inference and explicitly highlight discrepancies among methods that may warrant future research.

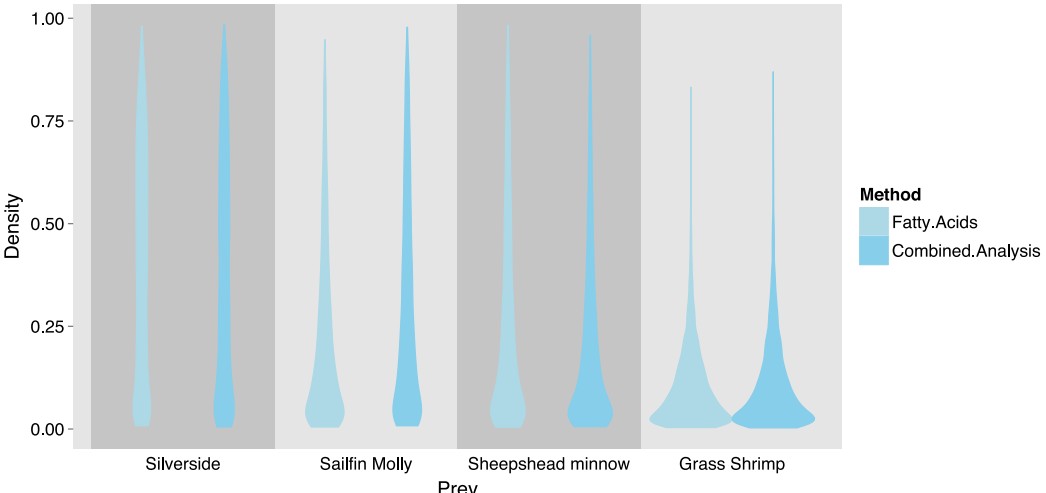

**Figure 6** **Posterior densities for diet proportion estimates of SF (fish only diet) treatment squid based on fatty acid (FA) profiles and a combined (FA & stable isotopes) analysis.** Note that no separate analysis using stable isotopes only was run.

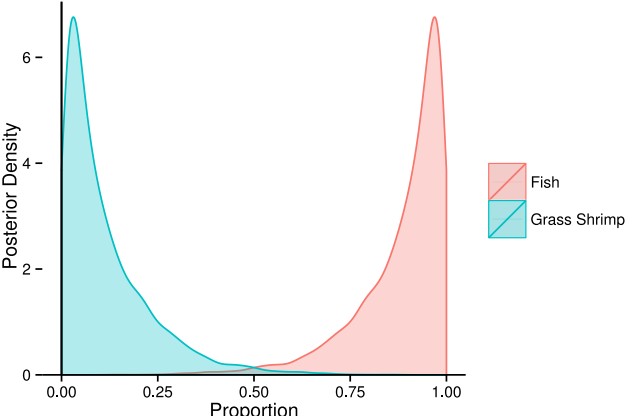

**Figure 7** **Posterior densities for diet proportion estimates of SF (fish only diet) treatment squid using using both fatty acid (FA) profiles and stable isotopes (SI), combining all fish species into a fish prey group.**

Simulation experiments and sensitivity tests suggested that the mixing model for FA profiles can achieve high accuracy of estimated diet proportions in idealized settings, and the application to squid feeding trials demonstrated the applicability of the model in a practical, albeit controlled setting. Our results in the squid study further confirm many of the points made by *Stowasser et al. (2006)*. In particular, our analysis of switched diet treatments suggested that despite the short acclimation time (10–15 days) we can detect dominant proportions of the switched diet treatments from both SI and FA. While a complete discussion of these findings is beyond the scope of this manuscript, these results suggest that the time frame over which FA profiles and SI integrate diet proportions in squid digestive glands is on the order of days to weeks rather than months.

Our results from the squid experimental data also highlighted the model sensitivities found using simulated data. Fish species within treatments could not be discriminated using FA profiles (and/or SI), and estimated diet proportions of the different fish species in the SF treatment remained very uncertain. This uncertainty reflects insufficient differences in fish prey FA profiles and SI to accurately discriminate among these different species within squid diets. Despite the uncertainty in estimated diet proportions for individual fish species, the estimate for the group of all fish species reveals a clear dominance of fish rather than crustaceans in the diets (Fig. 7). This example thus illustrates another benefit of a fully Bayesian treatment: rather than giving potentially misleading point estimates in such situations, the wide posterior distributions highlight the fact that there is insufficient signal in the data to discriminate among different fish species in the diets.

The decrease in accuracy with decreasing source separation reported from simulations and shown in the squid experiments is thus due to choosing a point estimate within a large interval rather than the model suggesting erroneous point estimates of diet proportions. Similarly, for unknown conversion coefficients, posterior distributions of diet estimates are generally wide, provided that the prior for conversion coefficients reflects uncertainty. Even when uncertainty about diet proportions is relatively low, posterior distributions of diet proportions close to 0 or 1 were generally skewed rather than symmetric due to the constrained nature of the diet proportions, meaning the posterior mode (the highest posterior probability) is often not located at the mean of the posterior distribution. In this case, as for very wide and/or flat posterior distributions, any point estimate chosen for diet proportions is somewhat arbitrary. Overall estimation error from (posterior mean) point estimates thus scales with the evenness of the diet proportions as well as overall uncertainty in diet proportions. Rather than relying on point estimates of diet proportions in that case, it becomes increasingly important to acknowledge uncertainty in the posterior distributions.

We opted for a fully Bayesian analysis that estimates prey and predator distributions, as well as individual proportions. However, the Bayesian approach for FA comes at a relatively high computational cost: we found that there are limits to the dimensionality that the estimation procedure (as we formulated it) can accomodate. When working with fully Bayesian methods in high dimensional applications such as FA profiles, where the number of measured variables can be large (>20 FAs is common), there is an inevitable trade-off between computational feasibility and model dimensionality. For instance, we have found that, in it's present form (V1.0), fastinR can handle a set of approximately 15 FAs for a set of 5 potential prey items, especially when MCMC chains are run in parallel (optional in the package). The software is thus currently not able to handle the large prey libraries that can be handeled in QFASA. Our aim is to further develop the fastinR package to include empirical Bayes options (as described in *Parnell et al., 2013*) that would likely speed up the models considerably. However, the empirical Bayes approach comes at the cost of considering prey distribution parameters as known quantities, which may not be desirable with a small number of prey samples. Therefore, we further aim to swap out the current JAGS back-end for Bayesian estimation against custom MCMC code to allow for increased model complexity and faster run-times.

Since the model dimensionality depends at once on the number of prey items, predators and FAs in the analysis, we have found it to be useful to initially use predator FA profile (geometric) means or relatively few predator signatures to estimate a single population distribution. Once one has determined that the model can effectively estimate diet proportions given the data at hand and knowledge of conversion coefficients, the model can be re-run with a larger number of predators and/or FAs and, although time consuming, may provide additional insights. The squid diet example illustrates this strategy: we first estimated population level parameters for predators (although we used all predator signatures rather than their geometric mean), and then proceeded to more complex analyses of individual diet proportions.

To further address the issue of computational complexity, we presented an approach to variable selection for FA profiles. An optimal subset of variables is usually one that explains the bulk of among prey variance (represented by CAP axes), but eliminates FAs that only contribute minimally to separation among sources, and thus only add noise. As such, this variable selection approach is not limited to FA profiles, but may be of use with other diet markers or chemical tracers for which dimension reduction is desirable prior to analysis. In our squid application, we found that retaining only 4 FA was enough to explain nearly 75% of among source variance, and adding additional FA only added a small amount of signal for rapidly increasing co-linearity in prey signatures. While a limited number of FA may often be diagnostic of a particular prey type, it may not generally be the case that a small number of FA account for the bulk of the signal. The computational cost of high dimensional models in the Bayesian framework can be limiting in such instances, and the practical trade-off between model run-time and accuracy of estimated diet proportions will have to be considered.

Recent developments in SI mixing models have led to increasingly realistic models in terms of their error structure (*Hopkins & Ferguson, 2012*) and incorporation of relevant biology, such as time dependent diet proportions and SI signatures (*Parnell et al., 2013*). Given that our FA profile and combined FA profile and SI models employ the same general structure as these models, such developments are achievable within this framework. It should be noted, however, that they present the practitioner with requirements for substantial amounts of data of various kinds (i.e., measurement error estimates, collection of SI and FA profiles through time, respectively), and may substantially increase computational requirements. Nevertheless, we suggest that the method presented here provides a basis to use and combine the two most powerful dietary markers available in a single framework to produce more robust and comparable diet estimation.

## ACKNOWLEDGEMENTS

The authors wish to thank Gabriele Stowasser for kindly preparing and sharing her data for re-analysis. The Jensen lab group, Finlay Thompson, Edward Abraham and Laureline Meynier provided helpful discussion that lead to various improvements in the software and the manuscript. This research was funded by the NOAA Northeast Cooperative Research Program. Findings and conclusions in this paper are those of the authors and do not

necessarily represent the views of the NMFS, NOAA. Reference to trade names does not imply endorsement by the National Marine Fisheries Service, NOAA.

### Funding

This research was funded by a grant from the NOAA Northeast Cooperative Research Program to the Garden State Seafood Association, Rutgers University, and Dragonfly Science (EA133F1). The funders had no role in study design, data collection and analysis, decision to publish, or preparation of the manuscript.

### Grant Disclosures

The following grant information was disclosed by the authors:
NOAA Northeast Cooperative Research Program: EA133F1.

### Competing Interests

Philipp Neubauer is an employee of Dragonfly Science.

### Author Contributions

- Philipp Neubauer conceived and designed the experiments, performed the experiments, analyzed the data, contributed reagents/materials/analysis tools, wrote the paper, prepared figures and/or tables, reviewed drafts of the paper, wrote R package.
- Olaf P. Jensen conceived and designed the experiments, contributed reagents/materials/analysis tools, wrote the paper, reviewed drafts of the paper.

### Data Deposition

The following information was supplied regarding the deposition of related data:
Neubauer, Philipp (2014): Squid fatty acids and Stable Isotope data. Figshare: http://dx.doi.org/10.6084/m9.figshare.1056245.

### Supplemental Information

Supplemental information for this article can be found online at http://dx.doi.org/10.7717/peerj.920#supplemental-information.

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
