# Peer review of "Bayesian estimation of predator diet composition from fatty acids and stable isotopes"

_PeerJ, doi:10.7717/peerj.920_

## Round 0.1 · original submission · Minor Revisions

Both reviewers make suggestions for improvement of the manuscript which seem relevant. Can you please address these comments?

·

Basic reporting

The submission seems to adhere to PeerJ policies: it is written in a clear style and provides the necessary background to contextualize the problem.

Experimental design

I see nothing wrong with the chosen methodology. The research question is well stated and explored. The authors clearly made a significant effort to ensure reproducibility of the experiments and results.

Validity of the findings

Data used is either public and/or generated by a function provided by the authors. From personal experience, provided code (https://github.com/Philipp-Neubauer/fastinR) is well documented and seems to work as described.

Additional comments

Some suggestions/comments:

1) On the first paragraph of the Methods and on page 4, the authors state that SI vectors are assumed to follow a multivariate normal distribution. Given that these values constitute ratios, this assumption seems reasonable only under particular circumstances (e.g. denominator with a low relative dispersion compared to numerator). Though I trust the authors to have specific reasons for this choice, it could be useful to provide a source or some comment to help support it, if possible.

2) Though the advantages of the authors' approach compared to other possible methods are clearly stated, it could be interesting to perform a side-by-side comparison of the performance of the proposed approach against e.g. QFASA, in terms of estimation errors regarding prey proportions (in the case of simulation data, where the ground truth is known). Quantifying the "advantage" of using the proposed method would be a good support to the theoretical arguments provided.

3) The authors suggest the use of a particular feature selection approach (based on Constrained Analysis of Principal components and then using the sum of the loadings scaled by the respective eigenvalues as a measure of variable importance) to limit computational cost to reasonable levels. Though this seems a valid approach, it could be interesting to assess whether using another feature selection approach would have any effect on estimation performance.

Minor corrections:

- On page 3 of the manuscript, perhaps the meaning of "clr" (in the term "clr transformations") should be also stated as "centered log ratio" (to be consistent with the clarification of the meaning of "alr transformation", on page 2).

- On the third line of the S3 document ("Selecting fatty acids for diet analysis: an ordination approach"), where it says "en empirical Bayes method", it should probably say "an empirical Bayes method".

Reviewer 2 ·

Basic reporting

Working in the area of estimation of predator diet composition, the authors present a Bayesian mixing model approach that uses fatty acid profiles (FAP) as well as a joint model that integrates analysis of both stable isotopes (SI) and FAP. The authors also provide an accompanying open source software package for R.

Overall, I found the manuscript to be well-organized and written in a clear manner. In my opinion the research was well motivated in the introduction and deals with a relevant and important problem in statistical ecology.

Minor Corrections:
p.1 , fourth line under INTRODUCTION: Replace “particularly” with “particular”.
p. 5 and p. 6: Should the same title be used under METHODS and RESULTS as the heading for the real-life application?
p. 7, Figure 2: Write out what “CAP” stands for.
p. 7, last line: “S” missing from “S5”.

Experimental design

While the Bayesian framework presented here is more complex than QFASA, the only other method available to estimate predator diet composition from FAs, as the authors point out, it allows explicit representation of uncertainty in the mixing proportions. I agree that being able to incorporate the various sources of uncertainty is important, however it appears that the authors may have missed the following contributions in this area:

Stewart C, Field C (2011) Managing the essential zeros in quantitative fatty acid signature analysis. J Agric Biol Environ Stat 16(1):45–69
Stewart C (2013) Zero-inflated beta distribution for modeling the proportions in quantitative fatty acid signature analysis. J Appl Stat 40(5):985–992

At the very least, I feel that these papers should be referenced since they provide confidence interval methodology based on QFASA for the diet proportions.

Another valuable aspect of the authors’ work is the development of a variable selection method for selecting FAs to use in the analysis. I surmise that this procedure would be useful to biologists using FAs to study the diet of predators, regardless of which diet estimation method they are using.

Validity of the findings

The authors use both simulations and real-life data to illustrate and validate their methods. They provide extensive details on their methods in supplemental material, in addition to an R package (called fastinR), and this material should allow other researchers to relatively easily apply the methods to their data.

Although I like the idea of providing the lengthy details of the simulation study in supplemental material, I would have liked to have seen a summary of some of this on p. 4 under “Simulation studies”. For instance, how many different prey species were used? What sample sizes were used? How many FAs were used? What were the true diet compositions used in the simulations?

While I think the authors do a decent job at describing the limitations of their work (for example, one limitation being, “the computational cost of high dimensional models”), I think it would be beneficial if they were able to be a bit more specific concerning limits on the model dimensionality in practice with respect to the number prey items, predators and FAs. QFASA is able to handle prey databases with 30 or so species and a large number of FAs (see above references, for example) and I could not get a sense as to whether it would be feasible to apply the Bayesian methodology to such applications or not.

---

## Round 0.2 · accepted · Accept

I believe you have adequately addressed the suggestions / minor criticism you received from both reviewers.